# Magnetically treated water for removal of surface contamination by Malathion on Chinese Kale (*Brassica oleracea* L.)

**Chadapust J. SUDSIRI**[1], **Nattawat JUMPA**[2], **Raymond J. RITCHIE**[3]*

**1** Faculty of Sciences and Industrial Technology Prince of Songkla University in Suratthani, Suratthani, Thailand, **2** Sciences Laboratory and Equipment Centre, Prince of Songkla University in Suratthani, Suratthani, Thailand, **3** Biotechnology of Electromechanics Research Unit, Faculty of Technology and Environment, Prince of Songkla University in Phuket, Phuket, Thailand

* raymond.r@phuket.psu.ac.th, raymond.ritchie@alumni.sydney.edu.au

**Data Availability Statement:** All relevant data are within the manuscript and its Supporting information files. The attachments are the datafiles for Figures 3 to 6. We have provided all the GC data

## Abstract

Malathion® is a persistent organophosphate pesticide used against biting and chewing insects on vegetables. It is a difficult-to-remove surface contaminant of vegetables and contaminates surface and ground water and soils. Malathion® is only partially water soluble, but use of detergent carriers makes adhering Malathion® residues difficult to subsequently remove. Magnetically treated water (MTW) successfully removed Malathion® from Chinese Kale (*Brassica oleracea* L.), meeting Maximum Residue Load (MRL) standards. Samples were soaked in MTW for 30 min prior to detection with GC/MS/MS, 98.5±3.02% of Malathion® was removed after washing by MTW. Removal by simple washing was only ≈42 ±1.2% which was not nearly sufficient to meet MRL criteria.

## Introduction

Food which is free of pesticide residues a great concern for the consumer. The high demand for vegetables requires use of fertilizers and pesticides to increase product yield, improve quality, and extend storage life [1]. Indiscriminate use of pesticide may result in toxicity [2] or unmarketable agricultural produce. The health hazard from residues in crops to farmworkers has also become a global problem.

Malathion® which is an organophosphorus (OP) pesticide (technically phosphotriesters) is used globally since it has a high efficiency in killing insects, particularly chewing insects [3]. Organophosphate insecticides like Malathion® have a common generic structure, with the phosphotriesters group being responsible for its insecticidal properties (Fig 1) and it has weak electrolyte chemical properties [4]. Like most organophosphates, Malathion® is only slightly soluble in water (145 mg L$^{-1}$ at 25°C; [5]) and its weak electrolyte properties make it detectable by conductivity in solution. Malathion® has no chlorinated hydrocarbon groups and is more readily biodegradable than chlorinated organophosphates such as Chlorpyrifos® [3, 5–8]. Organophosphate pesticides are potent nerve agents, functioning by inhibiting the action of acetylcholinesterase (AChE) in nerve cells [9, 10]. Their extensive use has raised increasing

sets as an Excel (r) file as requested. The data sets can be used to back-calculate our data presented in the paper.

**Funding:** Ch. J. S. is grateful for a stipend and financial support for this project from Prince of Songkla University—Suratthani (Research Development Grant SIT6001026S). The funders had no role in the study design, data collection and analysis, decision to publish or preparation of the manuscript.

**Competing interests:** The authors have declared that no competing interests exist.

environmental concern and they are risks to human health [11, 12]. Toxicity of Malathion[R] depends on metabolic activation; symptoms may appear from a few minutes to a few hours after exposure [10]. The microbial breakdown products of organophosphates in themselves can be medically dangerous [13]. Malathion[R] is more vulnerable to microbial breakdown [2] and is less persistent [2, 14] than chlorinated organophosphates [7]. Organophosphates have a reputation for rapid breakdown in the environment but this is not necessarily positive. In general, they are more slowly degradable than usually supposed and psychologically their reputation for ready degradability actually has the negative effect of encouraging careless use of organophosphates [8, 14].

Many methods have been considered to decontaminate foodstuffs of pesticides but some are not practical in restaurant/household application [15–22]. Simple washing by water is commonly recommended for households and restaurants [15–17]. Rinsing vegetables with tap water decreases levels in vegetables (but not necessarily sufficiently to pass health regulations) of several insecticides [18]. Since Malathion[R] is only poorly water soluble [5], Malathion[R] is generally applied in a formulation including a detergent carrier additive. An unfortunate consequence is that tap water is unable to effectively remove Malathion[R] residue because the residues persist as a recalcitrant deposit on the epidermis of plants after the dispersant has disappeared. Washing in 0.9% NaCl, 0.1% NaHCO$_3$, 0.1% acetic acid, 0.001% KMnO$_4$ and boiling have been tried [15–17, 19] but boiling is inappropriate for vegetables normally served

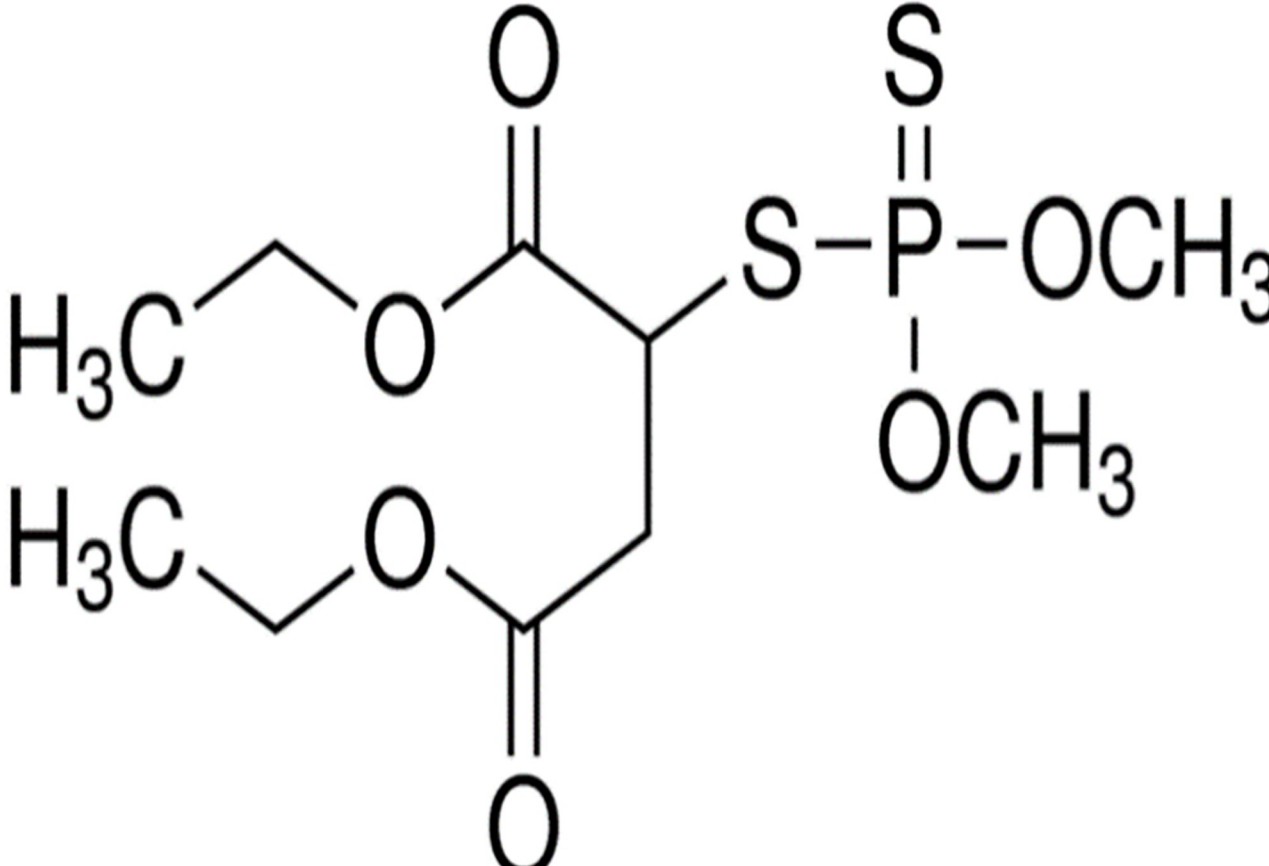

**Fig 1. Malathion[R] structure.** (https://en.wikipedia.org/wiki/Malathion#/media/File:Malathion.png).

uncooked [19]. The removal efficiency depends on physiochemical properties of the pesticide, degree of binding to plant surfaces, specificity of the chemical used as the removal agent and the carriers or dispersants used in application of the pesticide. Pesticide removal/degradation strategies such as gamma irradiation and natural sunlight [20], oxidation by ozone [21] and combinational use of $O_3/UV/TiO_2$ [22] have also been used but are not practical in restaurant/ household situations but might be useable in packaging in factories.

Water is a paramagnetic material. Recently we have demonstrated that Magnetically Treated Water (MTW) could be effectively used to remove the chlorinated organophosphate, Chloropyrifos [8], but Chlorpyrifos has been withdrawn from use in many countries. Malathion is still widely used and seems a suitable replacement for Chlorpyrifos. Researchers have worked on the influence of magnetic field (MF) on properties of MTW [23, 24]. Magnetically Treated Water (MTW) can remove scaling of metallic surfaces: this seems to be an important clue in its ability to remove recalcitrant materials adhering to biological material [22, 25]. Magnetization has measurable effects on pH of MTW (becomes more alkaline), dissolved oxygen, enhances conductivity and mineral solubility, increasing total water hardness and changes abiotic and biological calcification [26–29]. The MTW reduces surface tension [11, 28] and decreases its hydrophobicity due to clustering structure of water and improved polarizing effects [30]. MTW has been widely used in industry and construction owing to the convenient changes in the physicochemical properties of water, particularly interactions of water with surfaces. This appears to be the mode of action of MTW in removing Chlorphyrifos from the surfaces of vegetable [8]. MTW has known effects on seed germination, plant growth and development which appear to be largely surface and membrane permeability effects [31–33].

Malathion is routinely applied with a carrier dispersant or detergent, unfortunately, a consequence is that once the dispersant is removed the remaining residue deposited on vegetables may be very difficult to remove [18, 19]. Development of methods to remove or breakdown pesticides is a food science priority. The rationale for extensive banning of Chlorpyrifos has been largely based on its poor biodegradability and environmental impact [7, 34] rather than the issue of its removability from food stuffs. More attention should be paid to removal of organophosphates from vegetables because they are not as readily degradable as usually thought [2, 8, 14]. New pesticide removal methods are needed that are not technically demanding or potentially hazardous in themselves. Magnetic Treated Water (MTW) offers a convenient and importantly, a non-destructive method, of removal of adhering pesticides from fresh vegetables. MTW has been successfully applied to removal of the very water insoluble organophosphate, Chloropyrifos [8] but Chlorpyrifos is now banned in many countries: in this study we extend this to the much more widely used Malathion which is the logical choice to replace Chloropyrifos were biting and chewing insects are targeted. Unfortunately, the composition of detergent/dispersants used to apply pesticides are generally regarded as Intellectual Property (IP) and their identity is usually not available on commercial products.

## Materials and methods

An upgraded version of the device described by Surendran et al. [31] and Sudsiri et al. [8] was used as a generator of magnetically treated water (MTW) (Fig 2). The magnetisation equipment used for the investigation contained three (3) permanent magnet rods having magnetic intensity of 800–1200 mT in the piping for generating the magnetic field. The neodymium magnets ($Nd_2Fe_{14}B$), 300 mm in length and 25.4 mm in diameter, were from MagnetDD, 11/8 Moo.5 Plai Bang, Bang Kruai District, Nonthaburi, Thailand. The magnetic field strength was measured by telemeter (PHYWE, No. 13610–93, Göttingen, Germany). Domestic Tap Water

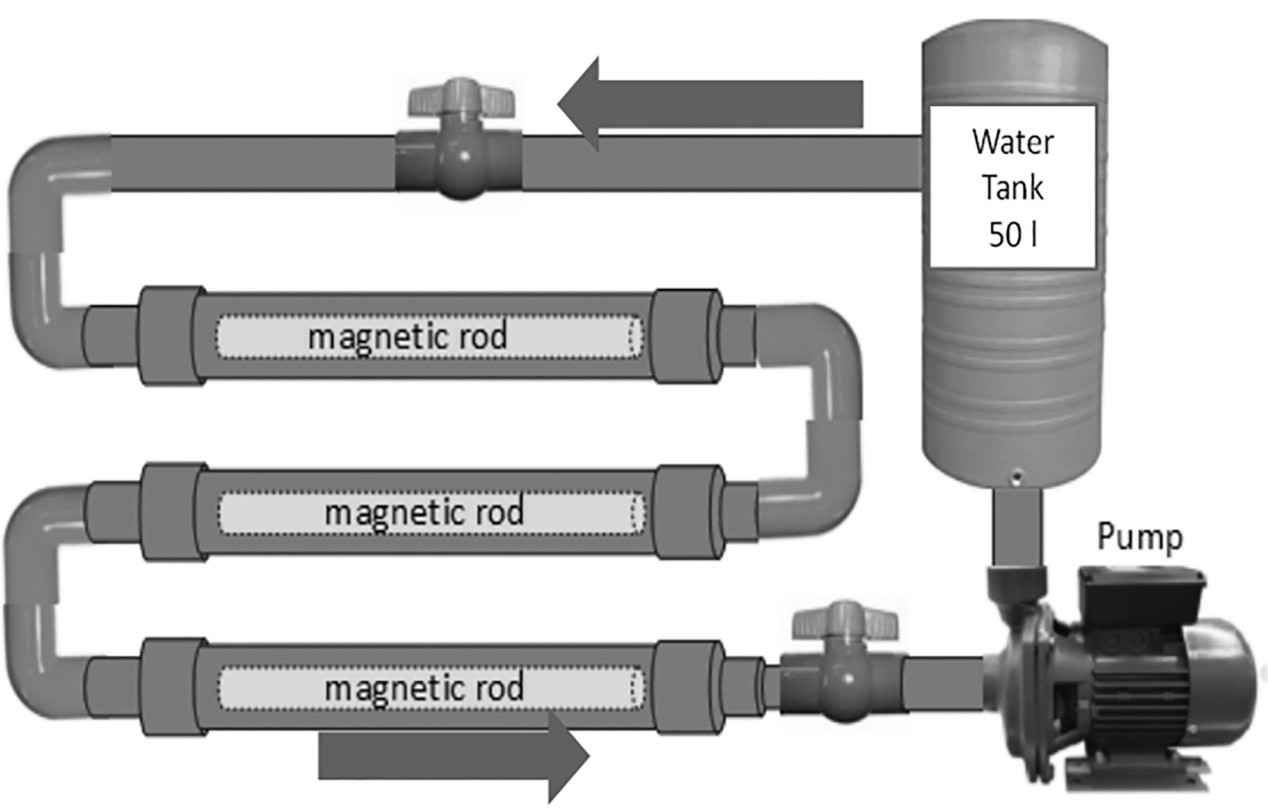

**Fig 2. Improved version of magnetic device for creating MTW.**

(TW) whose electric conductivity (EC) was ≈86 µS cm$^{-1}$ (8.6 mS m$^{-1}$) measured using a conductivity meter (Mettler Toledo MC,126-2M, Greifensee, Switzerland) and with a pH of about 7.8 (pH meter, Model pH 700, Eutech Instruments, Singapore) was used to prepare MTW. Experimentally, water was passed in a cycle continuously through the device through PVC pipes at the rate of 2 L s$^{-1}$ through the magnetizing apparatus and a ½ HP pump moved water into a 50 L plastic carboy storage tank to be recycled. After the magnetic treatment, the EC was recorded because magnetization causes a characteristic increase in EC of freshwater that is easy to monitor. Once a batch of water was magnetized (less than 1 h) the magnetization lasted for at least 24 h regardless of the magnetisation method (permanent magnets or electromagnets) [8, 31–33].

To enable this study to be relevant to actual field use the Malathion® used was a standard commercial product (40% w/v emulsifier concentration (EC)) (*Thaion Agro Chemical* CO., LTD. Yannawa, Bangkok, Thailand). Chinese Kale (*Brassica oleracea* L.) were germinated from seed in a greenhouse at a temperature about 30 °C. and 65% humidity. Plots consisting of 10 rows separated by 3 rows of Brassicaoleracea L. + 2 walkways (total distance ≥ 5 m to avoid cross contamination). After a grow-out period (70 days) half of each row was allocated to a randomized complete design with 3 replicates. The group without application of any pesticides was the designated control. Prior to pesticide application at 70 days (marketable size), the controls were moved into the row well away (> 5 m) from those plots chosen randomly for application of the pesticide. Care was taken not to contaminate the controls with pesticide during application. The vegetables for pesticide application were treated with Malathion® at the

**Table 1. Comparison of component area and their concentrations of extracted samples washed by tapwater (TW) and magnetically treated water (MTW) comparing to control (no pesticide contamination).** Zar [50] was used as the standard statistical reference text. Based on three replications (n = 3, means ± standard errors) of independent vegetable samples [50]. The concentrations were compared to Minimum Residual Load (MRL) on a pass/fail basis. The MRL for Malathion® is 3.000 mg L$^{-1}$ [39]. The controls had no detectable Malathion® but the sprayed vegetables failed the MRL criterion (p ≪ 0.001, t-test compared to guideline) [50]. Only the control vegetables and the Malathion®-sprayed vegetable washed with MTW passed MRL criteria based on students t-test compared to the guideline threshold value (p ≪ 0.001) [50].

| Samples | Component Area | Final Concentration (mg L$^{-1}$) | % removal | Level of MRL Criterion (Pass/Fail) |
|---|---|---|---|---|
| Control no pesticide treatment | Not detectable | zero | - | MRL Pass |
| No washing | 18,242,242±187882 | 7.040±0.116 | 0±0.00 | MRL Fail |
| Washed with Tapwater (TW) | 10,626,293±60743 | 4.142±0.134 | 41.74±1.230 | MRL Fail |
| Washed with MTW | 159,477±3189 | 0.157±0.0325 | 98.49±3.020 | MRL Pass |

recommended concentration of 2% w/v [21]. To ensure good coverage of the pesticide applications were carried out every 2 days for 1 week. After pesticide treatment, treatments were left 1-day prior to running residue analysis. To avoid cross contamination, each sample was collected in clean transparent air tight polyethylene bag and was properly labelled with sample number before further subjected to washing experiments.

Samples treated with Malathion® were washed to compare effectiveness of household/restaurant treatments (tap water wash) [17] and a MTW wash routine similar to those used previously in our study of removal of Chlorpyrifos® from a similar leafy vegetable [8]. Washing by tap water was carried out on approximately 1 kg samples (in triplicate) of harvested vegetables by immersion in tap water for 30 min. Washing with MTW was carried out by soaking ≈1 kg of samples in 5 L of MTW for 30 min. All washed samples were kept on blotting paper to remove excess water. More vigorous washing, multiple washing or cutting up the vegetable into very small pieces was not used so as to simulate actual household and restaurant practice. Malathion® was analysed using an Agilent 7890B GC combined with a 7000 D triple quadrupole MS (QQQ) operated in MRM mode (Agilent Technologies, Santa Clara, CA). The analysis of pesticide residues was carried out using the pesticide multi-residue QuEChERS (Quick Easy Cheap Effective Rugged and Safe) method as described by accepted methods, BS EN 15662 (E) [35–37] used previously [3, 8, 16, 17]. Wu et al. [38] used a method similar to QuEChERS to assay Chlorpyrifos®. A known standard of Malathion® was used for calibration of the GC (see S1 Fig) (Office of Scientific Instruments and Testing, Prince of Songkla University): the standard is traceable to the US Environmental Protection Agency Pesticide Repository (Fort Meade, MD, USA). Extraction of Malathion® was performed by extracting 10 g of homogenized Chinese Kale in 10 mL of acetone, with 15 mL anhydrous acetonitrile, 4 g of anhydrous magnesium sulphate, 1 g of sodium chloride and 1 g of anhydrous sodium acetate. This extraction process was followed by a cleaning procedure by transferring the supernatant (1 mL) into another tube containing 25 mg of primary-secondary amines (PSA) and 25 mg alumina N. After shaking and centrifugation for 5 min at 3000g, the extract supernatant was transferred to an auto sampler vial for GC-MS/MS analysis. Each analysis was for an independent sample (n = 3) of the vegetable material. Means and ± standard errors (SE) are quoted. The GC-MS/MS data sets are included as supplementary data files.

## Results

Analyses were carried out in triplicate (Table 1, see supplementary material for data sets). An example of a GC-MS/MS chromatogram of blank Chinese Kale extract is shown in Fig 3. No peaks identifiable as Malathion® were observed in the sample of Chinese Kale extract from the control plants. A chromatogram of an extraction from Chinese Kale exposed to Malathion® was constructed (Fig 4). There was a large, easily identified, peak with occupied an area of

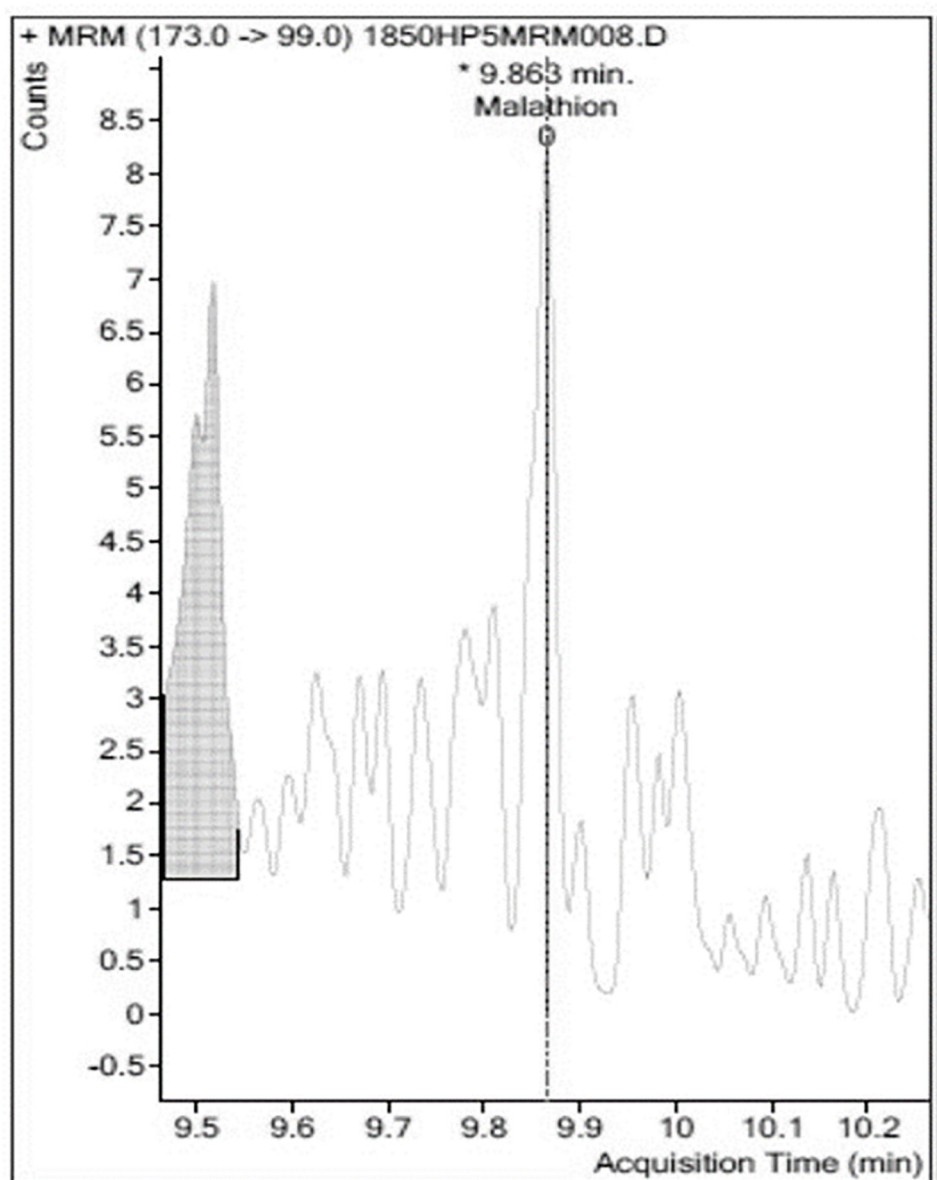

**Fig 3. Chromatogram of an example of a blank extraction of *Brassica oleracea* L. sample eluted by GC-MS/MS.** No detectable Malathion[®] was found in three blank runs.

18,242,242 which co-eluted with known Malathion [®]. A chromatogram of Chinese Kale washed by tapwater (TW) after extraction had a peak area of 10,626,293 (Fig 5). A chromatogram of an extracted sample from Chinese Kale washed by MTW is shown in Fig 6 (peak area of 159,477). Concentrations of Malathion[®] equivalent to the peak areas varied greatly (Figs 4–6, Table 1) and consequently the calculated percent removal from the samples varied greatly over a wide range (Table 1). The calculated concentrations in the vegetables were compared to the Maximum Residue Load (MRL), the highest level of a pesticide residue legally tolerated in, or on, vegetables when a pesticide has been applied. The MRL of Malathion[®] is 3.000 mg L$^{-1}$ [5, 39] and so the unwashed vegetables sprayed with Malathion[®] failed MRL criteria by a wide

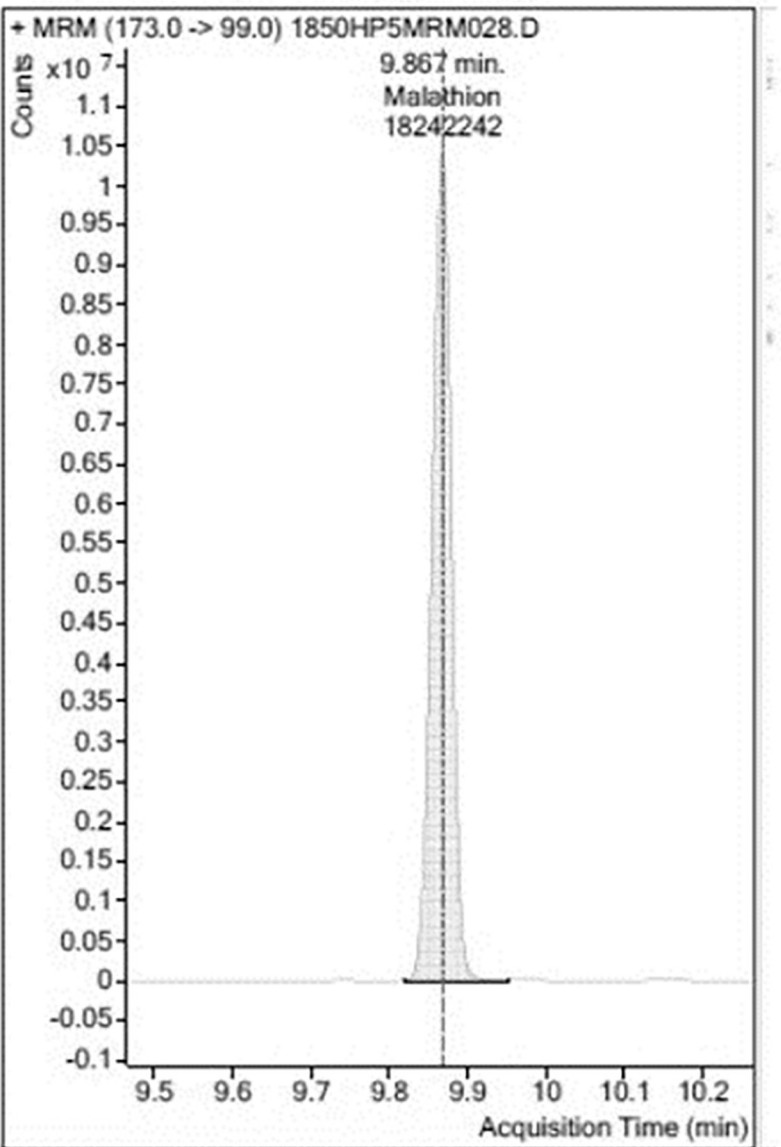

**Fig 4. Selected example of a chromatogram of *Brassica oleracea* L. containing Malathion® extract without washing peak area is 18,242,242 (7.0399775 mg L⁻¹).** The *Brassica* is heavily contaminated. All three replicates were heavily contaminated.

margin. Magnetically Treated Water (MTW) efficiently removed Malathion® contamination up to 98.49±3.02% (mean ± SE, n = 3) in Chinese Kale and consequently passed MRL criteria (MRL Pass); washing by tap water removed only 41.74±1.23% (mean ± SE, n = 3) of the Malathion®. The Malathion contamination was too high to pass the MRL guideline threshold.

## Discussion

This study has shown the efficiency of MTW to effectively remove the organophosphate insecticide Malathion® from fresh leaves *Brassica oleracea* L. deliberately contaminated with Malathion®. The results are closely comparable to our previous findings on Chlorpyrifos® which is

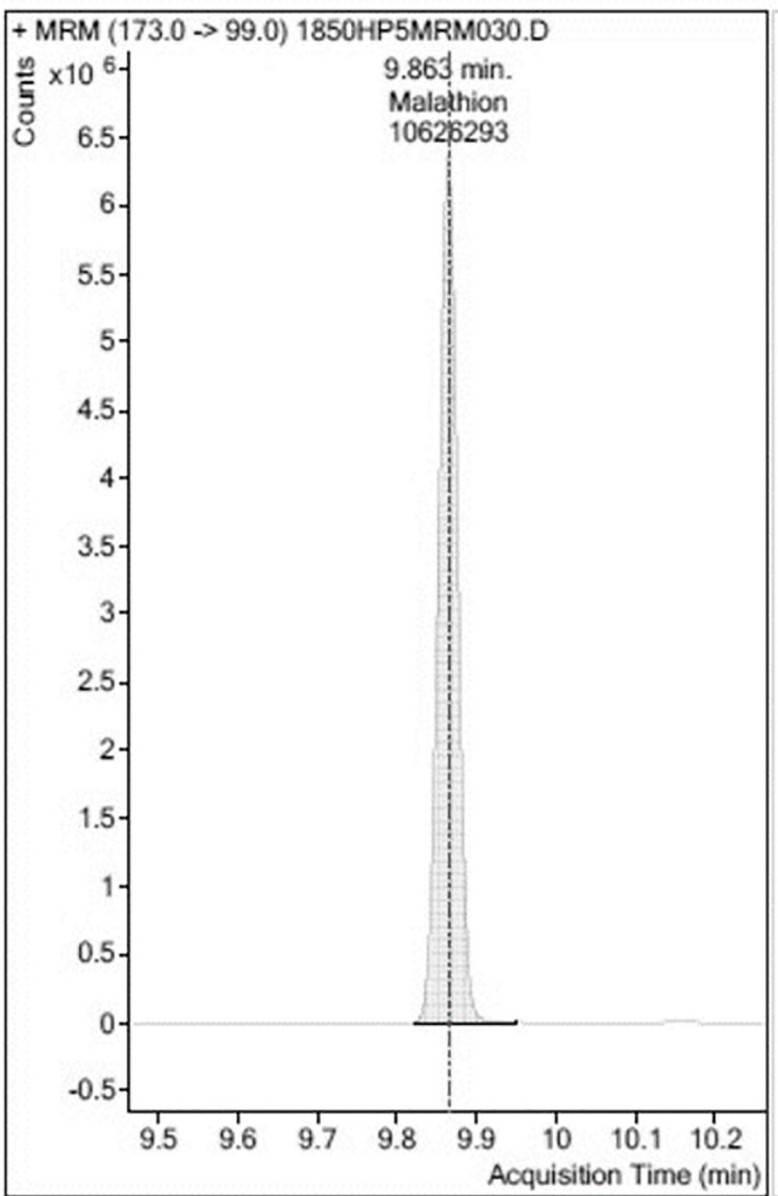

**Fig 5. A selected example of a chromatogram of an extract of *Brassica oleracea* L. washed by tapwater (TW).** The elution provides peak area is 10,626,293 (4.1411803 mg L$^{-1}$). Tapwater was not very effective in removing Malathion® (3 replicates).

much less soluble and is increasingly being withdrawn from use in many countries [8] whereas Malathion® continues to be widely used. Simple washing by tap water does not achieve an acceptable level of decontamination (Table 1) on vegetable material based upon the accepted MRL criteria [5, 19, 37, 39]. Washing *Brassica oleracea* L. using tap water can remove ≈ 42% of Malathion® compared to that of the controls with no washing. The results agree with Wang et al. [40] who studied decontamination of vegetables of Malathion® by washing with tap water and showed that simple washing with tapwater was not an effective nor reliable removal agent (unacceptably high variance and poor reproducibility). In this study, we used a simple

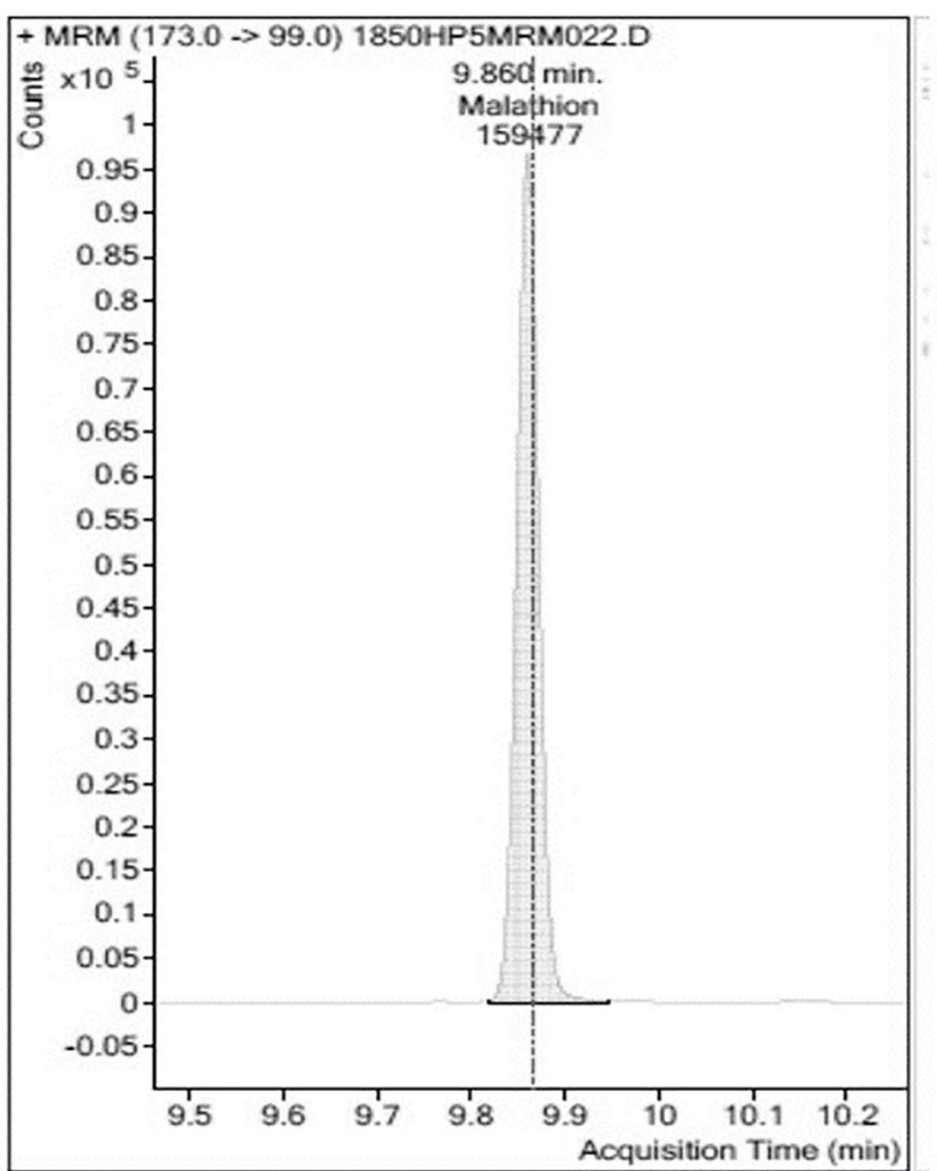

**Fig 6. A selected example of a chromatogram of extract from *Brassica oleracea* L. washed by Magnetically-Treated-Water (MTW).** The elution peak area is 159,477 (0.1572810 mg L$^{-1}$). MTW was very effective in removing most Malathion® but a trace remained (three replicates).

30 min soaking routine as a minimalist protocol to best represent how MTW treatment would be used in realistic restaurant and household use [15–19, 41].

Our washing of Chinese Kale with MTW removed about 98.5% of applied Malathion® (Table 1), reducing the herbicide residual to a value which easily passed MRL safety criteria (3.000 mg L$^{-1}$) [5, 39]. This is a much more effective level of decontamination compared to previous reports such as various washing solutions (0.9% NaCl, 0.1% NaHCO$_3$, 0.1% acetic acid, 0.001% KMnO$_4$, 0.1% ascorbic acid, 0.1% malic acid and 0.1% oxalic acid) [19, 37, 38], electrolysis [38] or ozone [40].

The mechanism of magnetization of water and its interactions are not yet fully understood theoretically [24, 27, 28, 41–47] making it difficult to formulate a theory of its mechanisms of action [8]. Our previous study on Chlorpyrifos [R] showed that it can be efficiently removed by MTW [8]. Possible explanations of our findings here that Malathion[R] can also be efficiently removed by MTW include magnetisation effects increasing the solubility of hydrophobic polar compounds [2, 42], reduced surface tension [2, 23, 42], a general effect on dissolved solids due to clustering of water molecules [45], improved polarizing effects [2, 30] and general improvements in organic matter solubility [41, 48]. Better knowledge of the properties of magnetized water and its mechanism of action [23, 47] especially in biological systems [49] are potentially valuable in finding novel practical applications: the lack of an adequate model for its mechanism-of-action should not discourage the impirical application of the technology [50].

## Supporting information

**S1 Fig. Calibration of Agilent 7890B GC/ 7000 D triple quadrupole MS with known standard Malathion [R].** The calibration on authentic malathion is included in the S5 File.
(TIF)

**S1 File. This file is the GC-MS data set used to prepare Fig 3.**
(PDF)

**S2 File. This file is the GC-MS data set used to prepare Fig 4.**
(PDF)

**S3 File. This file was for the tapwater treated plants and was used to prepare Fig 5.**
(PDF)

**S4 File. This file is the GC-MS data set from the magnetic water treated plants used to prepare Fig 6.**
(PDF)

**S5 File. This file is the complete set of EXCEL [R] files downloaded from the Agilent 7890B GC combined with a 7000 D triple quadrupole MS (QQQ) operated in MRM mode (Agilent Technologies, Santa Clara, CA).** The file includes the calibration using authentic Malathion.
(XLSX)

## Acknowledgments

The authors acknowledge the Office of Scientific Instruments and Testing, Prince of Songkla University for analysis of the pesticide samples using their facilities and expertise. The authors thank the Office for access to their certified Malathion [R] standard traceable to the US Environmental Protection Agency Pesticide Repository (Fort Meade, MD, USA).

## Author Contributions

**Conceptualization:** Chadapust J. SUDSIRI, Raymond J. RITCHIE.

**Data curation:** Chadapust J. SUDSIRI, Nattawat JUMPA.

**Formal analysis:** Chadapust J. SUDSIRI, Nattawat JUMPA, Raymond J. RITCHIE.

**Funding acquisition:** Chadapust J. SUDSIRI.

**Investigation:** Chadapust J. SUDSIRI.

**Methodology:** Chadapust J. SUDSIRI, Nattawat JUMPA, Raymond J. RITCHIE.

**Project administration:** Chadapust J. SUDSIRI.

**Supervision:** Chadapust J. SUDSIRI, Raymond J. RITCHIE.

**Visualization:** Raymond J. RITCHIE.

**Writing – original draft:** Chadapust J. SUDSIRI, Raymond J. RITCHIE.

**Writing – review & editing:** Chadapust J. SUDSIRI, Raymond J. RITCHIE.

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
