## [Decision Letter · Decision Letter 0]

10 Jul 2023

PONE-D-23-11448Magnetically treated water for removal of surface contamination by Malathion® on Chinese KalePLOS ONE

Dear Dr. Ritchie,

Thank you for submitting your manuscript to PLOS ONE. After careful consideration, we feel that it has merit but does not fully meet PLOS ONE’s publication criteria as it currently stands. Therefore, we invite you to submit a revised version of the manuscript that addresses the points raised during the review process.

We look forward to receiving your revised manuscript.

Kind regards,

Abdul Rauf Shakoori

Academic Editor

PLOS ONE

Journal Requirements:

“This work was supported by Thailand Science Research and Innovation Fund (Thai 252 Government Initiative Program) and Prince of Songkla University, Suratthani Campus.”

 "Ch. J. S. is grateful for a stipend and financial support for this project from Prince of Songkla University - Suratthani (Research Development Grant SIT6001026S).”

“Ch. J. S. is grateful for a stipend and financial support for this project from Prince of Songkla University - Suratthani (Research Development Grant SIT6001026S).”

Reviewers' comments:

Reviewer's Responses to Questions

**Comments to the Author**

1. Is the manuscript technically sound, and do the data support the conclusions?

Reviewer #1: Yes

Reviewer #2: Yes

2. Has the statistical analysis been performed appropriately and rigorously? 

Reviewer #1: No

Reviewer #2: Yes

3. Have the authors made all data underlying the findings in their manuscript fully available?

Reviewer #1: No

Reviewer #2: Yes

4. Is the manuscript presented in an intelligible fashion and written in standard English?

Reviewer #1: Yes

Reviewer #2: Yes

5. Review Comments to the Author

Reviewer #1: The research is valuable but the clear elucidation charts need to attached and statistics of comparison need to be more comparable and clear to support the idea and the results of the research although the elucidations will do that

Reviewer #2: In this paper entitled " Magnetically treated water for removal of surface contamination by Malathion® on Chinese Kale". This is an excellent manuscript, focusing on the problem of pesticide residues in agricultural products cleaning. The researchers demonstrated an effective technology, which can remove 98.5±3.02% Malathion®. I would be grateful if the researchers could answer the following questions:

1、Line58-62

It is suggested that Line58-62 be merged with the contents of the previous paragraph and rewritten into one paragraph.

2、Line 63, " Many methods have been considered to decontaminate foodstuffs of pesticides."

Please supplement this sentence with examples.

4、Line 77, punctuation should be added after "Chlorpyrifos ® [9] ".

5、Line 84, " [2730] " should be replaced by [27,30]. Please check this point all through the text.

6、Line86-88, "MTW has been widely used in industry and construction owing to the convenient changes in the physicochemical properties of water, particularly interactions of water with surfaces. "

It is suggested to delete this and supplement the application of MTW in pesticide treatment.

7、Line 88-89, "MTW has known effects on seed 88 germination, plant growth and development[9,32,34]"

It is suggested to delete this sentence

8、Line94-97, "The rationale for extensive banning of Chlorpyrifos ® has been largely based on its poor biodegradability and environmental impact [7,35] rather than the issue of its removability from food stuffs. New pesticide removal methods are needed that are not technically demanding or potentially hazardous in themselves. "

It is suggested to rewrite this sentence, which is inconsistent before and after.

9、Line 102-103, "Dispersants are used to apply an insecticidal surface residue on the sprayed plants."

Please confirm whether there is any problem in the description of this sentence.

10、References, The author cited few references in recent 3 years, and suggested to supplement them.

12、Please improve the resolution of Figure 3~ Figure 6.

6. PLOS authors have the option to publish the peer review history of their article (what does this mean?). If published, this will include your full peer review and any attached files.

Reviewer #1: No

Reviewer #2: No

---

## [Author Response · Author response to Decision Letter 0]

18 Jan 2024

PONE-D-23-11448R1

Magnetically treated water for removal of surface contamination by Malathion® on Chinese Kale (Brassica oleracea L.)

Dr Raymond J. Ritchie

Dear Dr. Ritchie,

We've checked your submission and before we can proceed, we need you to address the following issues:

1. Please note that funding information should not appear in the Acknowledgments section/Funding section or Any other areas of your Manuscript. We will only publish funding information present in the Funding Statement section of the online submission form. Please

remove any funding-related text from the manuscript.

L219-239 All funding information has been removed from the Acknowledgements etc.

2. Please upload a Response to Reviewers letter which should include a point by point response to each of the points made by the Editor

and / or Reviewers. (This should be uploaded as a 'Response to Reviewers' file type.) Please follow this link for more information:

https://protect-au.mimecast.com/s/I7cHCvl1rKiED9oq6CQpjqF?domain=blogs.plos.org

This has already been done. See previous change list.

3. Please remove your figures/ from within your manuscript file, leaving only the individual TIFF/EPS image files. These will be automatically included in the reviewer’s PDF

All figures in the clean version of the text have been removed as instructed.

4. Your Data Availability statement currently reads:

"The attachments are the datafiles for figs 3 to 6."

Additionally, in your cover letter, you write:

"We have included the data sets as Supplementary material.

-Prompts (a) & (b) concerning data access. There is a difficulty with the Data Availability Statement. We the authors are happy to share our

data with colleagues however The Senior Author (Chadapust Sudsiri) informed me that the guidelines of the Research Development Grant

SIT6001026S does not allow us to deposit our data in a public repository."

And in the additional information section, you have written:

"All the data is available from the senior author (Chadapust J. SUDSIRI) upon request."

To get around this problem we have included in the Supplementary Material full sets of the GC-MS data used for the project.

Before we proceed, we’ll require some additional clarification to ensure your submission adheres to the PLOS ONE Data Availability policy

(https://protect-au.mimecast.com/s/7RH0CwV1vMfvDjXk1CqTXId?domain=journals.plos.org).

1) Please clarify whether your manuscript contains your minimal data set. The minimal data set is defined as the data set used to reach the conclusions drawn in the manuscript with related metadata and methods, and any additional data required to replicate the reported study findings in their entirety. This may include: a.) The values behind the means, standard deviations and other measures reported; b.) The

values used to build graphs; c.) The points extracted from images for analysis (https://protect-au.mimecast.com

/s/3fslCxngwOfL2lKDAcYJZSZ?domain=journals.plos.org).

We confirm that the supplementary material contains all the necessary data. 

2) Please confirm whether there are legal or ethical restrictions on sharing your data publicly.

Since we are providing all the necessary data as Supplementary Material we do not think there are any legal difficulties even though we are apparently not allowed to put the data in a repository.

3) If legal or ethical restrictions apply, please provide all necessary instructions and non-author contact information (preferably email) for a data access committee, ethics committee, or other institutional body that other researchers would require to request access to your data.

Note that it is not acceptable for an author to be the sole named individual responsible for ensuring data access.

We are providing data access by providing the data sets as Supplementary material.

4) If there are no legal or ethical restrictions on sharing your data publicly, please upload the minimal anonymized data set necessary to replicate your study findings as either Supporting Information files, or to a stable public repository. If you upload your data to a repository,

please also provide the relevant URLs, DOIs, or accession numbers for other researchers to access your data directly. For a list of recommended public repositories, please see https://protect-au.mimecast.com

/s/JWlWCyojxQT7l1EKMCRwDnm?domain=journals.plos.org.

Once we receive this information, we will update your data availability statement on your behalf.

Problem solved by inclusion of the data sets in the Supplementary material.

---

## [Editor Report · Decision Letter 1]

24 Jan 2024

Magnetically treated water for removal of surface contamination by Malathion® on Chinese Kale (Brassica oleracea L.)

PONE-D-23-11448R1

Dear Dr. Ritchie,

We’re pleased to inform you that your manuscript has been judged scientifically suitable for publication and will be formally accepted for publication once it meets all outstanding technical requirements.

Kind regards,

Abdul Rauf Shakoori

Academic Editor

PLOS ONE

---

## [Editor Report · Acceptance letter]

4 Apr 2024

PONE-D-23-11448R1 

PLOS ONE

Dear Dr. Ritchie, 

I'm pleased to inform you that your manuscript has been deemed suitable for publication in PLOS ONE. Congratulations! Your manuscript is now being handed over to our production team.

Kind regards, 

on behalf of

Dr. Abdul Rauf Shakoori 

Academic Editor

PLOS ONE